# PROBABILISTIC SAMPLING-ENHANCED TEMPORAL-SPATIAL GCN: A SCALABLE FRAMEWORK FOR TRANSACTION ANOMALY DETECTION IN ETHEREUM NETWORKS

## ABSTRACT

The rapid evolution of the Ethereum network necessitates sophisticated techniques to ensure its robustness against potential threats and to maintain transparency. While Graph Neural Networks (GNNs) have pioneered anomaly detection in such platforms, capturing the intricacies of both spatial and temporal transactional patterns has remained a challenge. This study presents a fusion of Graph Convolutional Networks (GCNs) with Temporal Random Walks (TRW) enhanced by probabilistic sampling to bridge this gap. Our approach, unlike traditional GCNs, leverages the strengths of TRW to discern complex temporal sequences in Ethereum transactions, thereby providing a more nuanced transaction anomaly detection mechanism. Preliminary evaluations demonstrate that our TRW-GCN framework substantially advances the performance metrics over conventional GCNs in detecting anomalies and transaction bursts. This research not only underscores the potential of temporal cues in Ethereum transactional data but also offers a scalable and effective methodology for ensuring the security and transparency of decentralized platforms. By harnessing both spatial relationships and time-based transactional sequences as node features, our model introduces an additional layer of granularity, making the detection process more robust and less prone to false positives. This work lays the foundation for future research aimed at optimizing and enhancing the transparency of blockchain technologies, and serves as a testament to the significance of considering both time and space dimensions in the ever-evolving landscape of the decentralized platforms.

## 1  INTRODUCTION

Graph Convolutional Networks (GCNs) have emerged as a transformative tool in the domain of graph-structured data representation. Their ability to encapsulate both local and global graph structures has paved the way for their application in diverse fields. However, as the scale and intricacy of graph data have surged, the efficient training of GCNs has become a paramount concern. Traditional training paradigms, although effective, are often encumbered by high computational and storage demands, especially when dealing with expansive graphs. The realm of GCN training has witnessed a burgeoning interest in sampling methods, particularly those rooted in probabilistic frameworks. Layer-wise sampling methods have been at the forefront of recent advancements. Chen et al. (2018) in their work on FastGCN, for instance, championed the cause of probabilistic sampling on independent nodes. Their approach was further nuanced by Huang et al. (2018) in AS-GCN, which introduced the concept of layer-dependent sampling, thereby adding another dimension to the sampling process. The work of Zou et al. (2019) on LADIES further embellished this domain by integrating importance sampling, a pivotal technique that adjusts the sampling distribution to enhance efficiency. While these methods have significantly advanced the field, the literature also underscores the importance of other sampling paradigms. For instance, the work of Chiang et al. (2019) on Cluster-GCN underscores the potential of efficient algorithms in training expansive GCNs. Similarly, Zeng et al. (2020) with GraphSAINT have carved a niche by focusing on graph sampling-based inductive learning. The landscape of GCN research is further enriched by contributions from scholars like Ying et al. (2018) and Xu et al. (2019), who have delved into the intricacies of graph neural networks and their potential applications. Liu et al. (2023) mentioned that most temporal graph learning methods

model current interactions by combining historical information over time, however, such methods merely consider the first-order temporal information leading to sub-optimal performance. To solve this issue, they proposed extracting both temporal and structural information to learn more informative node representations.

Also, the topic of anomaly detection in Blockchain has received considerable attention. For example, in Ethereum, the unexpected appearance of particular subgraphs has implied newly emerging malware (Xu and Livshits 2019). Anomaly detection in blockchain transaction networks is an emerging area of research in the cryptocurrency community (Lee et al. 2022). Given that the Ethereum network witnesses dynamically evolving transaction patterns, it becomes imperative to account for the temporal sequences and correlations of transactions. While some literature has touched upon temporal networks, there is a conspicuous absence of comprehensive research that deeply integrates TRW with GCNs, and probabilistic sampling, especially within the blockchain environment. Furthermore, the specific challenge of anomaly and transaction burst detection in the Ethereum network, which has massive implications for network security and fraud detection, has not been extensively explored using these combined methodologies. As Ethereum continues to grow and evolve, addressing such gap with an appropriate methodology becomes increasingly crucial to ensure the security, scalability, and robustness of the network. This research endeavors to bridge this gap, proposing a Probabilistic Sampling-Enhanced Temporal-Spatial GCN, aiming to offer a more holistic understanding of both the spatial and temporal dynamics in the Ethereum transaction data, with potential contributions as:

**Effectiveness of TRW in Ethereum**: If GCN with TRW detects anomalies more effectively, it suggests that temporal patterns are vital in identifying irregularities in the Ethereum network. This could mean detecting suspiciously timed transactions, identifying token 'pump and dump' patterns, or finding smart contracts that exhibit strange behavior in their execution patterns over time.

**Efficiency in Ethereum Network**: Given the large and ever-growing size of the Ethereum blockchain, any method that provides an efficient way to sample representative nodes without sacrificing accuracy is valuable. If GCN trained with TRW nodes can do this, it's a significant advantage. The proof will be mainly discussed in the appendix.

**Generalization in Ethereum**: Ethereum transactions can involve complex interactions between various addresses and smart contracts. If a standard GCN without TRW misses certain anomalies or has more false positives, it may indicate the need to consider temporal sequences in the data.

**Detecting Sophisticated Attacks**: In the realm of decentralized networks, there are sophisticated attacks and frauds that depend heavily on timing, such as front-running attacks. If GCN with TRW can detect such patterns more effectively, it can become a valuable tool in blockchain security. Although this is a potential contribution, but we leave this topic in security for our future study.

## 2 MODEL DESIGN

Graph Convolutional Networks (GCNs) are a pivotal neural network architecture crafted specifically for graph-structured data. Through the use of graph convolutional layers, we seamlessly aggregate information from neighboring nodes and edges to refine node embeddings. In enhancing this mechanism, we incorporate probabilistic sampling, which proves particularly adept in analyzing the vast Ethereum network (see the appendix for the complete proof). The incorporation of Temporal Random Walks (TRW) adds a rich layer to this framework. TRW captures the temporal sequences in Ethereum transactions and when combined with probabilistic sampling, not only focuses on nodes' spatial prominence but also considers the transactional chronology. Here, 'time' in TRW is conceptualized based on the sequence and timestamps of Ethereum transactions, leading to a dynamically evolving, time-sensitive representation of the network.

### 2.1 GRAPH CONVOLUTION OPERATION

Here, we intend to give an overview of the fundamental concepts and main steps involved in the derivation.

Graph Representation: A graph is represented as $G = (V, E)$, where $V$ is the set of nodes (vertices) and E is the set of edges connecting the nodes. Node Features: Each node $v_i$ in the graph is associated with a feature vector $x_i$, representing its attributes or characteristics. Aggregation: Aggregation is a process to combine the feature vectors of neighboring nodes to obtain a summary representation.

The aggregation operation is typically defined as a weighted sum of the feature vectors of neighboring nodes, using an adjacency matrix $A$ to capture graph connectivity:

$$h_i = Aggregate\left(x_j | j \in N(i)\right) = \sum_{j \in N(i)} A_{ij} x_j \tag{1}$$

where $h_i$ is the aggregated representation of node $i$, $N(i)$ is the set of neighbors of node $i$, and $A_{ij}$ represents the weight between node $i$ and node $j$.

GCNs leverage weight sharing across nodes, which allows the same parameters to be used for aggregation and transformation of each node in the graph. This weight sharing enables GCNs to generalize across different nodes and learn graph-level patterns. Parameterization involves learning the weight matrix $W$ that will be shared across all nodes. The aggregated features are then transformed using the shared weight matrix:

$$h_i = Activation\left(\text{W Aggregate}\left(x_j | j \in N(i)\right)\right) \tag{2}$$

To enable information propagation across multiple layers, the graph convolution operation is performed iteratively through multiple graph convolutional layers (GCLs). The output of one layer serves as the input to the next layer, allowing the propagation of information through the network. The node representations are updated iteratively layer by layer, allowing information from neighbors and their neighbors to be incorporated into the node features. By applying these formulas, GCLs enable information propagation in the blockchain network by aggregating and updating node representations. The parameters $W^l$ are learned during the training process to optimize the model's performance on a specific graph-based task. GCNs often consist of multiple layers, where each layer iteratively updates the node representations:

$$h_i^{(l)} = Activation\left(W^{(l)}\text{Aggregate}\left(h_j^{(l-1)} | j \in N(i)\right)\right) \tag{3}$$

Here, $h_i^{(l)}$ is the representation of node $i$ at layer $l$, and $h_j^{(l-1)}$ is the representation of neighboring node $j$ at the previous layer ($l$-$1$). The formula for a multi-layer graph convolutional operation in a GCN is based on aggregating information from neighboring nodes and applying multiple layers of transformations. The final layer is usually followed by a global pooling operation to obtain the graph-level representation. The pooled representation is then used to make predictions.

## 2.2 INCORPORATING TEMPORAL RANDOM WALK (TRW) INTO GCN

The TRW-enhanced GCN creates a multidimensional representation that captures both the structural intricacies and time-evolving patterns of transactions. Such an approach requires meticulous mathematical modeling to substantiate its efficacy, and exploring the depths of this amalgamation can reveal further insights into the temporal rhythms of the Ethereum network.

**Temporal Random Walk (TRW)**

Given a node $i$, the probability $Pij$ of moving to a neighboring node $j$ can be represented as:

$$P_{ij} = \frac{\omega_{ij}}{\sum_k \omega_{ik}} \tag{4}$$

where $\omega_{ij}$ is the weight of the edge between node $i$ and $j$, and the denominator is the sum of weights of all edges from node $i$. In a TRW, transition probabilities take into account temporal factors. Let's define the temporal transition matrix $T$ where each entry $T_{ij}$ indicates the transition probability from node $i$ to node $j$ based on temporal factors.

$$T_{ij} = \alpha \times A_{ij} + (1 - \alpha) \times f(t_{ij}) \tag{5}$$

Where:
- $A_{ij}$ is the original adjacency matrix's entry for nodes $i$ and $j$.
- $\alpha$ is a weighting parameter.
- $f_{ij}$ is a function of the temporal difference between node $i$ and node $j$.

Given this temporal transition matrix $T$, a normalized form $\widetilde{T}$ can be used for a GCN layer:

$$\widetilde{T} = \widetilde{D}_T^{-1} T \tag{6}$$

Where $\widetilde{D}_T$ is the diagonal degree matrix of $T$. To incorporate the TRW's temporal information into the GCN, we can modify the original GCN operation using $\widetilde{T}$ in place of $\widetilde{A}$ :

$$h^{(l+1)} = \sigma \left( \widetilde{D}_T^{-\frac{1}{2}} \widetilde{T} \widetilde{D}_T^{-\frac{1}{2}} h^{(l)} W^{(l)} \right) \tag{7}$$

## 2.3 EFFECT ON ANOMALY DETECTION

For anomaly detection, the final embeddings from a GCN (post TRW influence) should be more sensitive to recent behaviors and patterns. When these embeddings are passed to a classifier or clustering and scoring algorithms (like DBSCAN, OCSVM, ISOLATION FORST, and LOF), anomalies that are based on recent or time-sensitive behaviors are more likely to stand out. There is a limitation that is inherent in unsupervised anomaly detection, especially in domains like Ethereum transactions where a definitive ground truth may not be readily available. In our work, the term "anomaly" refers to patterns that are statistically uncommon or divergent from the norm based on the features learned by our model. These uncommon patterns, while not definitively erroneous, are of interest because they deviate from typical behavior. In the context of Ethereum transactions, such deviations could potentially indicate suspicious activities, novel transaction patterns, or transaction bursts.

In conclusion, while the above provides a mathematical insight, the true value of TRW in improving GCN for anomaly detection is empirical. We would need to compare the performance of GCN with and without TRW on a temporal dataset to see tangible benefits.

1. Node Features are Weighted by Time: When updating the node features through the matrix multiplication, nodes that are temporally closer influence each other more, allowing recent patterns to be highlighted.
2. Temporal Relationships are Captured: The modified node features inherently capture temporal relationships because they aggregate features from temporally relevant neighbors.
3. Higher Sensitivity to Recent Anomalies: With temporal weighting, anomalies that have occurred recently will be more pronounced in the node feature space.

**Theorem:** Enhanced Anomaly Detection in GCNs through Temporal Random Walk Integration.

**Proof.**

1. **Essence of Anomaly Detection:**
At a fundamental level, anomaly detection is the task of distinguishing outliers from normal data points in a given feature space. If we have an anomaly score function $s : \mathbb{R}^d \to \mathbb{R}$, we can detect anomalies by: $s(v) > \theta$ Where $\theta$ is a threshold, and $v$ is a vector in the feature space.

2. **GCN's Role:**
A GCN produces node embeddings (or features) by aggregating information from a node's neighbors in the graph. Let's express this aggregation for a single node using a simple form of a GCN layer:

$$h_i^{(l+1)} = \sigma \left( \sum_{j \epsilon Neighbors(i)} W h_j^{(l)} \right) \tag{8}$$

Where $h_i^{(l)}$ is the feature of node $i$ at layer $l$, and $W$ is the weight matrix.

3. **Incorporating TRW:**
With a time-aware random walk, the aggregation process is influenced by time, so the aggregation becomes:

$$h_i^{(l+1)} = \sigma \left( \sum_{j \epsilon Neighbors(i)} T_{ij} W h_j^{(l)} \right) \tag{9}$$

Where $T_{ij}$ is the temporal transition probability from node $j$ to node $i$. Let's assume a node with an anomaly will have a different feature vector from the nodes without anomalies. For simplicity, let's

use the Euclidean distance as the anomaly score: $s(v) = \|v - \mu\|$ where $\mu$ is the mean vector of all node features. Given a temporal anomaly (an anomaly that's influenced by recent events), using TRW will result in a modified feature vector for the anomalous node. Let's consider two scenarios:

1. GCN without TRW: For an anomalous node $n$, its feature vector is: $h_n = \sigma \left( \sum_j W h_j \right)$

2. GCN with TRW: For the same anomalous node $n$, it becomes: $h'_n = \sigma \left( \sum_j T_{nj} W h_j \right)$

If the anomaly is temporally influenced, then $h'_n$ should be significantly different from $h_n$ due to the weights introduced by $T_{nj}$ (see appendix A for weight cancellation). In the context of our anomaly score function: $s(h'_n) - s(h_n) > \delta$ where $\delta$ is a value indicating the sensitivity of the temporal context; we will use this later in our scoring method. If the anomaly is truly temporally influenced, this difference will be significant, and thus, the GCN with TRW will have a higher likelihood of detecting the anomaly. From the linear algebra perspective, the effect of TRW on a GCN for anomaly detection is evident in how node features are aggregated. The time-aware weights (from $T_{ij}$) make the GCN more sensitive to temporal influences, making it more adept at detecting anomalies.

**Theorem:** TRW sampling maintains higher temporal consistency than traditional random walk sampling.

**Definitions and Assumptions:**

- In TRW, the probability of transitioning from node $i$ at time $t$ to node $j$ at time $t + 1$ is given by $P_{ij}(t, t + 1)$, which is higher for temporally closer nodes.
- In a traditional random walk, the transition probability $P_{ij}$ is independent of time and is based solely on the adjacency matrix of the graph.
- Let $T_{\text{TRW}}(t)$ be the transition matrix for TRW at time $t$, where each entry $T_{ij}(t)$ represents the probability of transitioning from node $i$ to node $j$ at time $t$.
- Let $T_{\text{RW}}$ be the transition matrix for a traditional random walk, where each entry $T_{ij}$ is constant over time.
- Temporal consistency can be quantified by the variation in the transition probabilities over time. For TRW, this variation is expected to be lower than for traditional random walks, as TRW emphasizes temporal proximity.

**Proof.**
**Temporal Variation in TRW:**

- Consider the difference in transition probabilities between two consecutive time steps in TRW: $\|T_{\text{TRW}}(t+1) - T_{\text{TRW}}(t)\|$. This norm is expected to be small, indicating high temporal consistency.

**Temporal Variation in Traditional Random Walk:**

- For a traditional random walk, the transition probabilities do not change over time: $\|T_{\text{RW}}(t + 1) - T_{\text{RW}}(t)\| = 0$. However, this does not imply temporal consistency, as it does not account for the temporal nature of the data.

**Comparison:**

- To demonstrate higher temporal consistency in TRW, one can show that the variation in transition probabilities in TRW is more aligned with the temporal dynamics of the data compared to traditional random walk. This can be done by analyzing the correlation between $T_{\text{TRW}}(t)$ and the actual temporal sequence of events in the data.
- $T_{\text{TRW}}(t)$ aligns more closely with the temporal sequence of events than $T_{\text{RW}}$ and temporal consistency is better captured by a model that adjusts its transition probabilities based on the temporal proximity of events. Therefore, TRW is expected to maintain higher temporal consistency than traditional random walk sampling.

**Theorem:** Improvement of GCN performance with probabilistic sampling in the context of random walk sampling.

**Proof.** see appendix B for the complete proof.

## 3 EMPIRICAL ANALYSIS

GCNs have achieved state-of-the-art performance in various image recognition problems due to their ability to automatically learn hierarchical features from raw data. Researchers have recognized the potential of GCNs in tackling complex blockchain data analysis, particularly in Ethereum. The graph convolution operation combines the features of neighboring nodes to update the representation of a given node.

### 3.1 DATASETS AND DATA PREPROCESSING

This section discusses the publicly available Ethereum datasets and how we obtain them. Creating a complete transaction graph for all Ethereum blocks would be a computationally intensive task, as it would involve processing and storing a large amount of data. Additionally, the Ethereum blockchain is constantly growing, so the graph would continuously expand as new blocks are added. However, we provide an algorithm to generate a transaction history graph for a range of blocks. The following simple algorithm demonstrate how to create a graph of transactions between Ethereum addresses within a specified block range.

1. Place `<your_ethereum_node_url>` with the URL of the Ethereum node obtained from Infura/Alchemy website in order to access the Ethereum Mainnet.

2. Set the `start_block` and `end_block` variables to specify the block range to create the transaction history graph; one can create a descending range(`latest_block`, 0, -1).

3. The `create_transaction_history_graph` function obtains the transaction data.

We need to incorporate Temporal Node Features, and enhance the node features to capture temporal aspects more explicitly. We introduce time series data as features for each node to capture dynamic behavior over time. For instance, the number of transactions for an Ethereum address in a 10-day window could be used. This would change the feature representation for each node to a time series rather than a static feature vector. Others could include features like:

**activity_rate**: We can define this as the total number of transactions (both incoming and outgoing) of a node divided by the duration (in terms of blocks) for which we have information.
**change_in_activity**: The difference in the number of transactions of the current block with the previous block for a node. This would mean adding a record of the previous block's transaction count for every node.
**time_since_last**: The difference between the current block number and the block number of the last transaction involving that node.

### 3.2 TRW- GCN COMBINED METHOD TO DETECT ANOMALIES

To apply graph convolutional layers to the blockchain data for aggregating information from neighboring nodes and edges, we'll use the PyTorch Geometric library. This library is specifically designed for graph-based data and includes various graph neural network layers, including graph convolutional layers. Note that training and testing a graph neural network on Ethereum dataset would require significant computational resources, as currently, the Ethereum network possesses over 10 million blocks, which are connected over the Ethereum network. Here we provide the transaction history graph within a specified block range.
In Algorithm 1, we intend to compare the anomaly detection of full- and sub-graphs (sampling using TRW). The graph convolution operation combines the features of neighboring nodes to update the representation of a given node. As node features, we input the 7 features indicated in 3.1 as vector representation; considering 20 hidden layers, 100 epochs, and `lr=0.01`, the resulting output vector aggregates information from all neighboring nodes. Training and testing a graph neural network on the Ethereum dataset would require significant computational resources. By using the nodes from TRWs for training, the GCN will be more attuned to the time-dependent behaviors in the Ethereum network, leading to better detection of sudden spikes in transaction volume or unusual contract interactions that occur in quick succession. By sampling nodes with TRWs, one prioritizes nodes with recent temporal activity when training the model. Here's how it can be done:

| Algorithm 1: TRW- GCN combined method to detect anomalies |
| --- |

**Steps:**

1. Load and Preprocess the graph $G$.

2. Node feature extraction for each node $v_i \in V$: Construct a node feature matrix $F \in \mathbb{R}^{|V| \times 4}$ where each row $F_i$ corresponds to $f(v_i)$.

3. Convert graph to adjacency matrix $A \in \mathbb{R}^{|V| \times |V|}$.

4. Instantiate two GCN models $M_{TRW}$ and $M$ with parameters in_channels, hidden_channels, out_channels.

5. Temporal Random Walk (TRW) for $k = 1$ to num_walks: Aggregate all walks in a set $W = \bigcup_{k=1}^{\text{num\_walks}} w_k$.

6. Training using subgraphs: Train $M_{TRW}$ or $M$ using node features $F_N$ and adjacency matrix $A_N$.

7. Anomaly Detection: Apply DBSCAN, One-Class SVM, IsoForest, and LOF on embeddings from the trained GCN model $M$ to obtain anomaly labels.

1. **Perform TRWs to Sample Nodes for Training:** The TRWs provide sequences of nodes representing paths through the Ethereum network graph. Nodes appearing frequently in these walks are often involved in recent temporal interactions.

2. **Train the GCN with the Sampled Nodes:** Instead of using the entire Ethereum network graph for training, use nodes sampled from the TRWs. This approach tailors the GCN to recognize patterns from the most temporally active parts of the Ethereum network.

Using the GCN with TRW combined method, one can achieve 1) nomalies Detected, 2) Training Efficiency, and 3) Quality of Embeddings. The integration of Temporal Random Walk (TRW) with Graph Convolutional Networks (GCNs) offers a novel approach for generating embeddings that capture both spatial and temporal patterns within the Ethereum network. These embeddings are vital for understanding the underlying transaction dynamics and for effectively detecting anomalous activities. To evaluate the potential of the TRW-GCN methodology in the realm of anomaly detection, we employ four distinct machine learning techniques: DBSCAN, SVM, Isolation Forest (IsoForest), and Local Outlier Factor (LOF).

The extensive use of these four diverse techniques allows us to validate the efficacy of the TRW-GCN framework. The high anomaly detection rates in Figure 4 by clustering methods (DBSCAN and SVM) underscores the importance of algorithm selection, indicating a higher false positive rate and therefore, not appropriate ML methods. As observed in Figure 1 (left), 100 blocks, these techniques do not seem sensitive to the embeddings generated by TRW-GCN, as the number of anomalies detected are similar with and without TRW, because in 100 blocks temporal characteristic of transactions is not significant. This becomes obvious, when we compare this with Figure 1 (right), where we investigate anomalies for 1000 blocks. leading them to perceive many deviations as anomalies. It's essential to note that high detection doesn't necessarily imply high accuracy; it might indicate a higher false positive rate in clustering ML methods. Moreover, our comparative analysis depicted in Figure 1 (right) vividly showcases the superiority of the TRW-GCN combined approach over traditional GCN with higher anomaly detection for IsoForest and LOF methods (over 4000 anomalies detected vs 0). The enhanced detection capabilities can be attributed to the TRW's ability to encapsulate temporal sequences and correlations of transactions. It is more interesting to find out which node feature mainly contributes to anomaly detection, we show it in Figure 2. As ilustrated by different colors, the feature 3-6 namely incoming value count, activity rate, change in activity, time since last (mainly the temporal features) are the drivers of anomalies. By recognizing time-dependent patterns, TRW-GCN approach can detect sophisticated anomalies that might elude traditional methods, emphasizing the need for considering temporal patterns in anomaly detection.

### 3.3 SCORE-BASED ANOMALY PATTERN IN TERMS OF TIME-DEPENDENT BEHAVIORS

While traditional methods like IsoForest and LOF compute anomaly scores based on the relative position or density of data points in the feature space, we need a method to be more focused on temporal dynamics, tracking the evolution of each node's embedding over time and weighing it by the node's frequency in the graph. To adapt the code to pick up anomalous patterns associated with time-dependent behaviors, the algorithm should be equipped to recognize such patterns. Here's how we can achieve this:

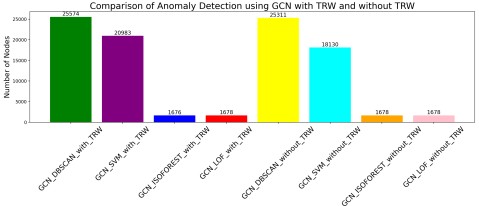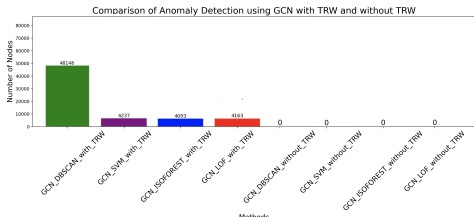

Figure 1: A comparison of 4 detection models namely dbscan, svm, isoforest and lof between full-graph and sub-graph with TRW sampling (left) 100 blocks (right) 1000 blocks which include 83252 nodes, and 101403 edges.

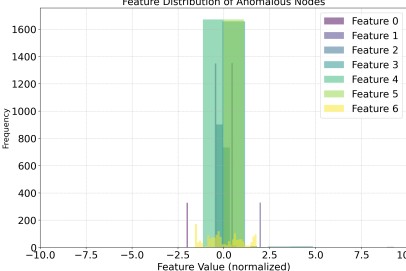

Figure 2: Feature distribution where anomalies detected: outgoing_tx_count, incoming_tx_count, outgoing_value_count, incoming_value_count, activity_rate, change_in_activity, time_since_last; Green colors: activity_rate, change_in_activity, and times_since_last show highest frequencies.

**Node Features Update:** Augment the node features to capture recent activity, as explained in datasets and preprocessing section.

**Anomaly Score Computation:** After obtaining node embeddings from the GCN, compute an anomaly score for each node based on its temporal behavior. The simplest way to achieve this is by computing the standard deviation of the node's feature over time and checking if the latest data point deviates significantly from its mean.

**Visual Representation:** Use the computed anomaly score to visualize nodes that are considered anomalous. Nodes with a higher anomaly score can be highlighted in the graph representation.

In this integrated code, Algorithm 2, we altered the node features to capture recent activity. After training the GCN and obtaining embeddings, we then compute an anomaly score based on how much the recent transaction volume (the latest day in our case) deviates from the mean. We then use a visualization function to display nodes with an anomaly score beyond a certain threshold (in this case, we've used a z-score threshold of 2.0 which represents roughly 95% confidence).

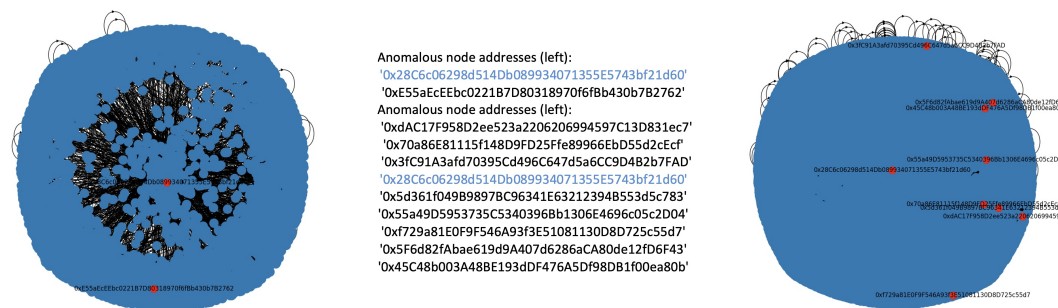

Figure 3: Comparison anomaly detection (left) 100 blocks with 8 features, (right) 1000 blocks with 4 features, (middle) the anomalous addresses where the common ones are marked blue.

In Figure 3, black points represent the vast majority of nodes in the Ethereum network dataset. They signify regular non-anomalous Ethereum addresses. Cluster of points inside and around the blue circle represent groupings of Ethereum addresses or contracts that have had frequent interactions with each other. The density or proximity of points to each other indicates how closely those addresses or contracts are related. Red points would represent the nodes that have been flagged as anomalous based on their recent behavior. The code identifies them by computing an anomaly score, and those exceeding a threshold are colored red. As easily observed, in the left graph, there are just

| Algorithm 2: A Score-based anomaly detection associated with time-dependent behaviors |
|---|
| **Graph Preprocessing:** $G' = G(V, E)$ where $E$ has node attributes. |
| **Node Feature Extraction:** $X = [x_1, x_2, \ldots, x_n]$ for $n \in V$. |
| **Adjacency Matrix:** $A$ from $G'$. |
| **GCN Model Definition:** GCNModel with layers: in_channels $\rightarrow$ hidden_channels $\rightarrow$ out_channels. |
| **Temporal Random Walk:** TRW$(G', \text{start}, \text{length})$ returns walk $W$ and timestamps $T$. |
| **Node Sampling via TRW:** All_Walks $= \bigcup_{i=1}^{\text{num\_walks}}$ TRW$(G', \text{random\_node}, \text{walk\_length})$. |
| **Node Frequency Computation:** $\text{freq}(v) = \frac{\text{occurrences of } v \text{ in All\_Walks}}{\text{max occurrences in All\_Walk}}$ for $v \in V$. |
| **Anomaly Score Computation:** $S(v) = \frac{(\text{emb}(v)_{\text{latest}} - \mu(\text{emb}(v)))}{\sigma(\text{emb}(v))} \times \text{freq}(v)$ where emb is the node embedding, $\mu$ is the mean, and $\sigma$ is the standard deviation. |
| **Visualization:** Highlight nodes $v$ where $S(v) > \text{threshold}$. |

2 nodes detected as anomaly in 100 blocks where we used 7 features in our detection algorithm (here we've used a z-score threshold of 2.0 (corresponding to 95% confidence); by changing the threshold, more/less anomalies could be detected), while in the right graph, we used 4 features to detect anomalies in 1000 blocks. There is one address in common between them, and one address different. We purposely did this to signify the importance of temporal feature selection. Therefore, by adding more tempooral features we would be able to detect missing anomlaies; which sounds a very promising method to detect temporal anomaly patterns.

### 3.4 HOW TEMPORAL RANDOM WALK (TRW) IMPACTS ON GCN PERFORMANCE

Let's delve into empirical justification on why TRW sampling could enhance the performance of GCNs, especially on temporal networks like Ethereum. For a detailed mathematical proof on the probabilistic sampling in GCN, you are invited to read appendix B. One issue with traditional random walks is the potential for creating "jumps" between temporally distant nodes, breaking the temporal consistency. GCNs rely on the local aggregation of information. Since TRW promotes smoother temporal signals, GCNs can potentially learn better node representations. Temporal consistency ensures that the sequences are logically and temporally ordered. This can be crucial for predicting future events or understanding time-evolving patterns, making GCNs more reliable.

Comparing different GCN models for fullgraph, and subgraph sampled with traditional and temporal random walk, in Figure 4, and seeing little difference between the accuracy of the fullgraph and the subgraphs, one can conclude Effective Sampling that traditional random walk and the temporal random walk are both effective in capturing the essential properties of the graph.

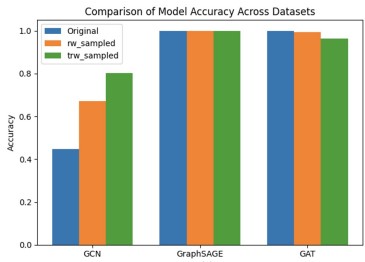
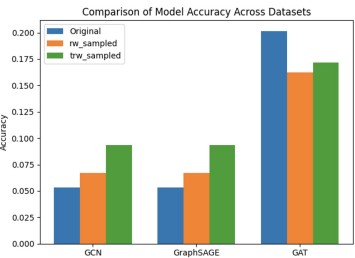

Figure 4: Comparison of accuracy of three GCN models between fullgraph, traditional RW and TRW-based subgraph in (left) 100 blocks (right) 1000 blocks; where the differences are bigger.

## 4 CONCLUSION

The evolution and complexity of the Ethereum network has heightened the urgency for robust anomaly detection methods. Through our research, we've demonstrated that the convergence of GCNs and TRW offers a solution to this challenge. This fusion has enabled us to delve deeper into the intricate spatial-temporal patterns of Ethereum transactions, offering a refined lens for anomaly detection. While this approach is used to obtain the embedding, we have compared different clustering and scoring methods to decrease false positives on anomalous activities. Not only we have demonstrated the analytical model in how GCN-TRW improves anomaly detection, but also in the appendix we proved analytically how probabilistic sampling could improve GCN performance.

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

## A  PROOF OF TEMPORAL INFLUENCE IN ANOMALY DETECTION

In the realm of anomaly detection, particularly in systems where temporal factors play a crucial role, the design and behavior of the transformation matrix $T_{nj}$ are of paramount importance. This section delves into the potential challenges posed by weight cancellation within $T_{nj}$ and its consequential impact on the detection of temporally influenced anomalies (given in our first Theorem). We explore the nuances of how such weight cancellations can diminish the efficacy of anomaly detection and propose strategies to mitigate these issues. Additionally, we underscore the importance of empirical analysis in validating the robustness and reliability of our anomaly detection methodology under various scenarios.

**Vector Spaces and Definitions**

Let $\mathbb{R}^m$ be the vector space of interest. Define $h_n \in \mathbb{R}^m$ as a feature vector in the absence of temporal influence. Let $T_{nj} \in \mathbb{R}^{m \times m}$ be a transformation matrix encoding temporal weights. Define $h'_n = T_{nj}h_n$, where $h'_n \in \mathbb{R}^m$ is the transformed feature vector under temporal influence.

Assume $T_{nj}$ has entries $t_{ij}$ where $i, j = 1, 2, \ldots, m$.

### DIFFERENCE MEASUREMENT

We use the Euclidean norm to quantify the difference: $||h'_n - h_n||_2$. This norm is given by

$$||h'_n - h_n||_2 = \sqrt{\sum_{i=1}^{m}(h'_{ni} - h_{ni})^2} \tag{10}$$

where $h'_{ni}$ and $h_{ni}$ are the components of $h'_n$ and $h_n$, respectively.

### EXPRESSION OF $h'_n$ IN TERMS OF $T_{nj}$ AND $h_n$

$h'_n = T_{nj}h_n$ implies

$$h'_{ni} = \sum_{j=1}^{m} t_{ij}h_{nj} \tag{11}$$

for each $i$.

### NORM CALCULATION

Compute the norm $||h'_n - h_n||_2$ as follows:

$$||h'_n - h_n||_2 = \sqrt{\sum_{i=1}^{m}\left(\sum_{j=1}^{m} t_{ij}h_{nj} - h_{ni}\right)^2}.$$

This equation represents the Euclidean norm of the difference between the transformed feature vector $h'_n$ and the original feature vector $h_n$.

### CONDITIONS FOR SIGNIFICANT DIFFERENCE

Given: $h'_n = T_{nj}h_n$ and $||h'_n - h_n||_2 > \epsilon$, for some threshold $\epsilon > 0$.

$$||h'_n - h_n||_2 = \sqrt{\sum_{i=1}^{m} \left( \sum_{j=1}^{m} t_{ij} h_{nj} - h_{ni} \right)^2}$$

For $||h'_n - h_n||_2 > \epsilon$, it must hold that $\sum_{i=1}^{m} \left( \sum_{j=1}^{m} t_{ij} h_{nj} - h_{ni} \right)^2 > \epsilon^2$.

This inequality implies that, for at least one $i$, the inner sum $\sum_{j=1}^{m} t_{ij} h_{nj} - h_{ni}$ must be non-negligible. Therefore, the weights in $T_{nj}$ must be such that they do not merely scale $h_{ni}$ but rather significantly alter the distribution of $h_n$. Scaling would imply a uniform change across all components of $h_n$, which might not be sufficient to meet the inequality condition. Instead, the transformation must significantly alter the distribution of $h_n$. This could mean changing the relative magnitudes of its components, modifying their relationships, or introducing non-linear changes. Such alterations are necessary for effectively differentiating between normal and anomalous states, especially in the context of anomaly detection where temporal influences are considered.

SCENARIOS LEADING TO WEIGHT CANCELLATION

**Scenario Analysis:**

Consider the case where $T_{nj}$ has symmetric properties or specific patterns that lead to cancellation.

For instance, if $T_{nj}$ is such that $t_{ij} = t_{ji}$ for all $i, j$, and $h_n$ has symmetric properties, then

$$\sum_{j=1}^{m} t_{ij} h_{nj} \text{ could approach } h_{ni} \text{ for all } i. \tag{12}$$

Additionally, if $T_{nj}$ contains complementary weights, such as some $t_{ij}$ and $t_{ik}$ summing to zero, and $h_{nj}$ and $h_{nk}$ are similar, cancellation could occur.

**Analysis of $T_{nj}$ Properties for Cancellation**

To further understand how $T_{nj}$ might lead to weight cancellation:

- Consider the spectral properties of $T_{nj}$. If $T_{nj}$ has eigenvalues close to 1, then it acts close to an identity matrix on certain vectors.
- If $T_{nj}$ has orthogonal rows or columns, it might preserve the magnitude of $h_n$ under certain conditions, leading to minimal change in $h'_n$.
- If the entries of $T_{nj}$ are structured such that they negate each other when applied to $h_n$, this could lead to a scenario where $h'_n \approx h_n$.

In scenarios where the weights in the transformation matrix $T_{nj}$ cancel each other out, this can significantly impact the detection of temporally influenced anomalies. To mitigate these issues, several strategies can be employed:

**Regularization:** Introducing regularization is a good practice in preventing extreme weight values, which can be beneficial in any anomaly detection system, including Ethereum network analysis.

**Weight Initialization and Optimization:** Carefully initializing the weights in $\mathbf{T_{nj}}$ and employing robust optimization techniques can ensure that the weights evolve in a manner that minimizes the risk of cancellation. This can be particularly important in Ethereum network anomaly detection.

**Spectral Analysis:** Performing spectral analysis of $\mathbf{T_{nj}}$ to understand its eigenvalues and eigenvectors can provide insights into how the matrix behaves and identify potential scenarios where cancellation might occur. Adjustments can then be made accordingly.

**Ensemble Methods:** Using ensemble methods is a robust strategy in anomaly detection, as it reduces reliance on any single transformation. In the context of Ethereum network anomalies, ensemble methods can enhance the reliability of detection by combining multiple models or transformations.

## B    IMPROVEMENT OF GCN PERFORMANCE WITH PROBABILISTIC SAMPLING

Providing a comprehensive mathematical proof on the theorem on improvement of GCN performance through probabilistic sampling in the context of analyzing the Ethereum network, even in a simplified scenario, is a complex task that requires careful consideration and detailed mathematical derivations.

Scenario: Consider a simplified Ethereum transaction graph with N accounts (nodes), and M transactions (edges) between them. We aim to prove the performance improvement of a GCN using probabilistic sampling for the task of predicting account behaviors.

Assumptions: 1. Nodes (accounts) have features represented by vectors in a feature matrix X.
2. The adjacency matrix A represents transaction relationships between accounts.
3. Binary labels Y indicate specific account behaviors.

**Proof.**

### B.1    DEFINE THE GRAPH LAPLACIAN

Start with the definition of the normalized graph Laplacian $L = I - D^{-\frac{1}{2}} A D^{-\frac{1}{2}}$, where $D$ is the diagonal degree matrix and $A$ is the adjacency matrix.

### B.2    TRADITIONAL GCN PERFORMANCE

Derive the eigenvalues and eigenvectors of the Laplacian matrix L and show their significance in capturing graph structure. Derive the performance of a GCN trained on the full graph using these eigenvalues and eigenvectors:

Step 1: Deriving Eigenvalues and Eigenvectors of the Laplacian matrix $L$

Given the normalized graph Laplacian matrix $L$, let $\lambda$ be an eigenvalue of $L$ and $v$ be the corresponding eigenvector. We have $L_v = \lambda_v$. Solving for $\lambda$ and $v$, we get:

$$D^{-\frac{1}{2}} A D^{-\frac{1}{2}} v = (1 - \lambda) v \tag{13}$$

$$A D^{-\frac{1}{2}} v = (1 - \lambda) D^{\frac{1}{2}} v \tag{14}$$

This equation implies that $D^{-\frac{1}{2}} A D^{-\frac{1}{2}}$ is a symmetric matrix that is diagonalized by the eigenvectors $v$ with corresponding eigenvalues $1 - \lambda$. The eigenvectors and eigenvalues of $L$ capture the graph's structural information. Larger eigenvalues correspond to well-connected clusters of nodes in the graph, while smaller eigenvalues correspond to isolated groups or individual nodes.

Step 2: Deriving GCN Performance Using Eigenvalues and Eigenvectors

Now let's consider a scenario where we're using a GCN to predict node labels (such as predicting high-value transactions) on the full graph. The GCN's propagation rule can be written as:

$$h^{(l+1)} = f(\hat{A} h^{(l)} W^{(l)}) \tag{15}$$

where $h^{(l)}$ is the node embedding matrix at layer $l$, $f$ is an activation function, and $\hat{A} = D^{-\frac{1}{2}} A D^{-\frac{1}{2}}$. is the symmetrically normalized adjacency matrix, and $W^{(l)}$ is the weight matrix at layer l. The key insight is that if we stack multiple GCN layers, the propagation rule becomes:

$$h^{(L)} = f(\hat{A} h^{(L-1)} W^{(L-1)}) = f(\hat{A} f(\hat{A} h^{(L-2)} W^{(L-2)}) W^{(L-1)}) \dots \tag{16}$$

We can simplify this as:

$$h^{(L)} = f\left( \hat{A}^{(l)} h^{(0)} W^{(0)} \prod_{l=1}^{L-1} W^{(l)} \right) \tag{17}$$

Using the spectral graph theory, we know that $\hat{A}^{(l)}$ captures information about the graph's structure up to L-length paths. The eigenvalues and eigenvectors of $\hat{A}^{(l)}$ indicate the influence of different subgraphs of length L on the node embeddings. Larger eigenvalues correspond to more significant

graph structures that can impact the quality of learned embeddings. By leveraging the spectral insights, GCNs can focus their learning on graph structures that matter the most for the given task. In the case of probabilistic sampling, the convergence of eigenvalues signifies that the sampled graph retains essential structural information from the full graph. This implies that by training GCNs on $\hat{A}_{\text{sampled}}$, we are effectively capturing the key graph structures necessary for accurate predictions. This incorporation of spectral properties aligns the GCN's learning process with the inherent characteristics of the graph, resulting in improved performance. The embeddings learned by the GCN on the sampled graph become more indicative of the full graph's properties as the sample size increases, enabling more accurate predictions or more efficient training convergence.

## B.3 PROBABILISTIC SAMPLING APPROACH

In this step, we'll introduce a probabilistic sampling strategy to select a subset of nodes and their associated transactions. This strategy aims to prioritize nodes with certain characteristics or properties, such as high transaction activity or potential involvement in high-value transactions. Assign a probability $p_i$ to each node i based on certain characteristics. For example, nodes with higher transaction activity, larger balances, or more connections might be assigned higher probabilities. For each node i, perform a random sampling with probability $p_i$ to determine whether the node is included in the sampled subset. Consider a graph with $N$ nodes represented as $N = \{1, 2, \ldots, N\}$. Each node i has associated characteristics described by a feature vector $\mathbf{X}_i = [X_{i,1}, X_{i,2}, \ldots, X_{i,k}]$, where $K$ is the number of characteristics. Define the probability $p_i$ for node i as a function of its feature vector $\mathbf{X}_i$: $p_i = f(\mathbf{X}_i)$. Here, $f(\cdot)$ is a function that captures how the characteristics of node i are transformed into a probability. The specific form of $f(\cdot)$ depends on the characteristics and the desired probabilistic behavior. For example, $f(\mathbf{X}_i)$ could be defined as a linear combination of the elements in $\mathbf{X}_i$:

$$p_i = \sum_{j=1}^{K} \omega_j X_{i,j} \tag{18}$$

Where $\omega_j$ are weights associated with each characteristic. The weights $\omega_j$ can be used to emphasize or downplay the importance of specific characteristics in determining the probability. After obtaining $p_i$ values for all nodes, normalize them to ensure they sum up to 1:

$$p_{\text{normalized}} = \frac{p_i}{\sum_{j=1}^{N} p_j} \tag{19}$$

Use the normalized probabilities $p_{\text{normalized}}$ to perform probabilistic sampling. Nodes with higher normalized probabilities are more likely to be included in the sampled subset, capturing the characteristics of interest. The specific form of $f(\cdot)$ and the choice of weights $\omega_j$ depend on the nature of the characteristics and the goals of the analysis. This approach allows for targeted sampling of nodes that exhibit desired characteristics in a graph.

## B.4 GRAPH LAPLACIAN FOR SAMPLE GRAPH

Given the sampled adjacency matrix $\hat{A}_{\text{sampled}}$, we want to derive the graph Laplacian $\hat{L}_{\text{sampled}}$ for the sampled graph. The graph Laplacian $\hat{L}_{\text{sampled}}$ is given by:

$$\hat{L}_{\text{sampled}} = I - \hat{D}_{\text{sampled}}^{-\frac{1}{2}} \hat{A}_{\text{sampled}} \hat{D}_{\text{sampled}}^{-\frac{1}{2}} \tag{20}$$

Where $\hat{D}_{\text{sampled}}$ is the diagonal degree matrix of the sampled graph, where each entry dii corresponds to the degree of node i in the sampled graph, and $\hat{A}_{\text{sampled}}$ is the sampled adjacency matrix.

$$d_{ii} = \sum_{j=1}^{N_{\text{sampled}}} \hat{A}_{\text{sampled},ij} \tag{21}$$

The modified Laplacian captures the structural properties of the sampled graph and is essential for understanding its graph-based properties.

We derived

$$\hat{L}_{\text{sampled}} = I - \hat{D}_{\text{sampled}}^{-\frac{1}{2}} \hat{A}_{\text{sampled}} \hat{D}_{\text{sampled}}^{-\frac{1}{2}} \tag{22}$$

as the normalized graph Laplacian for the sampled graph. Let $\hat{\lambda}_i$ be the $i$-th eigenvalue of $\hat{L}_{\text{sampled}}$ and $\hat{v}_i$ be the corresponding eigenvector. We have

$$\hat{L}_{\text{sampled}} \hat{v}_i = \hat{\lambda}_i \hat{v}_i \tag{23}$$

The goal is to compare the eigenvalues of $L$ with the eigenvalues of $\hat{L}_{\text{sampled}}$ and show convergence under certain conditions.

**Theoretical Argument:**
As the sample size $N_{\text{sampled}}$ approaches the total number of nodes $N$ in the original graph, $\hat{L}_{\text{sampled}}$ converges to $L$. This implies that the eigenvalues of $\hat{L}_{\text{sampled}}$ converge to the eigenvalues of $L$.

1. **Step-wise convergence:**

   For simplicity, we'll denote the entries of $\hat{L}_{\text{sampled}}$ as $\hat{l}_{\text{sampled}}(i,j)$ and the entries of $L$ as $l(i,j)$. To prove the convergence, we need to show that $\hat{l}_{\text{sampled}}(i,j) \to l(i,j)$ as $N_{\text{sampled}} \to N$ for all $i$ and $j$.

2. **Eigenvalue convergence:**

   Once establishing that each entry of $\hat{L}_{\text{sampled}}$ converges to the corresponding entry of $L$, one can use this result to prove the convergence of eigenvalues. Eigenvalues are solutions to the characteristic equation of the matrix, which depends on its entries. If all entries of $\hat{L}_{\text{sampled}}$ converge to those of $L$, the characteristic equations of both matrices will be similar.

**Stochastic Convergence:** The convergence argument relies on the concept of stochastic convergence. As the sample size becomes large, the sampled graph's properties approach those of the original graph. This includes the behavior of the eigenvalues.

**Graph Structure Alignment:** The convergence occurs when the sampled subset of nodes is representative enough of the entire graph. This means that the sampled graph captures the structural characteristics that contribute to the eigenvalues of L. Under the assumption of sufficient representativeness and with a large enough sample size, the eigenvalues of L^sampled converge to the eigenvalues of L.

## B.6 CONVERGENCE OF GCN EMBEDDINGS

Recall that the graph convolutional operation can be expressed as

$$h^{(l+1)} = f(\hat{A} h^{(l)} W^{(l)}) \tag{24}$$

The spectral properties of $\hat{A}$ and $L$ are determined by the eigenvalues. As shown in the previous steps, as the sample size increases, the eigenvalues of $\hat{L}_{\text{sampled}}$ converge to those of $L$. Graph convolutional layers rely on the eigenvectors and eigenvalues of $\hat{A}$. The graph convolution operation $\hat{A} h^{(l)} W^{(l)}$ involves these spectral properties.

**Convergence of GCN Layers:** Because the eigenvalues of $\hat{A}$ and $L$ are converging, the impact of multiple graph convolutional layers on $h$ and $h_{\text{sampled}}$ becomes increasingly similar as the sample size increases.

1. **Layer-by-layer impact:**

   As we stack multiple graph convolutional layers, each layer applies the graph convolution operation sequentially. This means that the impact of each layer depends on the eigenvalues of $\hat{A}$.

2. **Convergence influence:**

   As the eigenvalues of $\hat{A}$ converge to those of $L$ due to the increasing sample size, the behavior of the graph convolutional layers on $h$ and $h_{\text{sampled}}$ becomes increasingly similar.

The convergence of eigenvalues indicates that the structural characteristics of the sampled graph are aligning with those of the original graph. The graph convolutional layers are sensitive to these structural properties, and as the structural properties become more aligned, the impact of these layers on embeddings $h$ and $h_{\text{sampled}}$ will also become more aligned.

Since the top eigenvectors correspond to the major variations in the graph, as the spectral properties converge, the embeddings $h$ and $h_{\text{sampled}}$ learned by the GCN should increasingly align in terms of the top eigenvectors. As the eigenvalues of the Laplacian matrices converge, the behavior of the graph convolution operation and the resulting embeddings in both the original and sampled graphs becomes more similar. This implies that the embeddings learned by a GCN on the sampled graph $h_{\text{sampled}}$ will converge to the embeddings learned on the full graph $h$.

### B.7    Impact of Eigenvector Alignment on GCN Performance

Recall that the eigenvalues and eigenvectors of the Laplacian matrix capture the graph's structural properties. Eigenvectors corresponding to larger eigenvalues capture important patterns and variations in the graph.

**GCN Performance Analysis:**

1. **Predictive Power of Eigenvectors:** The alignment of top eigenvectors suggests that the information captured by these eigenvectors is consistent between the original and sampled graphs.

2. **Prediction Task:** If the prediction task relies on features that align with the graph's structural patterns, then the embeddings learned on the sampled graph will capture similar patterns as those on the full graph.

3. **Performance Convergence:** As the embeddings $h_{\text{sampled}}$ approach $h$ in terms of the top eigenvectors, the predictive performance of the GCN on the sampled graph should approach the performance on the full graph. The alignment of top eigenvectors implies that the information encoded in the embeddings learned by a GCN on the sampled graph converges to that of the embeddings learned on the full graph. This suggests that as $h_{\text{sampled}}$ converges to $h$, the predictive performance of the GCN on the sampled graph should approach that on the full graph, assuming the prediction task is influenced by the graph's structural patterns captured by these eigenvectors. However, the precise impact will depend on the nature of the graph, the quality of the sampling strategy, and the specific prediction task. To prove the improvement of GCN performance with probabilistic sampling, consider the following steps:

    (a) **Original Graph Performance (Without Sampling):** Let $E$ be the performance measure (e.g., accuracy) of the GCN trained and evaluated on the full graph $G$ using embeddings $h$, denoted as $E_{\text{full}}$.

    (b) **Sampled Graph Performance (With Probabilistic Sampling):** Now, consider the performance of the GCN trained and evaluated on the sampled graph $G_{\text{sampled}}$ using embeddings $h_{\text{sampled}}$, denoted as $E_{\text{sampled}}$.

    (c) **Improved Performance:** $E_{\text{sampled}} > E_{\text{full}}$ indicates an improvement in performance due to probabilistic sampling.
        - Utilize the previously shown argument: As the sample size increases, the embeddings $h_{\text{sampled}}$ converge to $h$ in terms of top eigenvectors.
        - With the alignment of top eigenvectors and the graph convolutional layers' convergence, the learned embeddings become more similar.
        - The improved alignment of embeddings captures more relevant structural information, potentially leading to improved prediction accuracy or other performance metrics.

By leveraging the convergence of embeddings and the improved alignment of top eigenvectors through probabilistic sampling, we can argue that the performance of the GCN on the sampled graph $G_{\text{sampled}}$ is expected to be better (higher accuracy, faster convergence, etc.) than on the full graph $G$. This proof highlights the positive impact of probabilistic sampling on enhancing the performance of GCNs in analyzing complex graphs like the Ethereum network.

