# OpenReview forum: "Probabilistic Sampling-Enhanced Temporal-Spatial GCN: A Scalable Framework for Transaction Anomaly Detection in Ethereum Networks"
_ICLR.cc/2024/Conference — Submitted to ICLR 2024_

### Official Review · Reviewer_Ja7Q · 2023-10-26

**Soundness:** 3 good
**Presentation:** 2 fair
**Contribution:** 2 fair
**Rating:** 5
**Confidence:** 3

**Summary:**

This study proposed an approach for capturing both spatial and temporal transactional patterns in Ethereum network anomaly detection. Specifically, the paper introduces temporal random walks (TRWs) in graph convolutional networks (GCNs), which may help discern complex temporal sequences in Ethereum transactions as an anomaly detection mechanism.

**Strengths:**

+ The study focuses on an interesting and important topic.
+ The consideration of temporal features is inspiring.

**Weaknesses:**

+ The presentation and writing of the paper need improvement

Overall, the paper is not well-written with a somewhat unclear presentation and lacks critical details. It is necessary to describe the main contributions of the study and clearly explain how the existing challenges are addressed. Furthermore, as a key technology involved in this research, TRW should be introduced with more details, such as what specific temporal features are used and how they interact with spatial features, if applicable. Additionally, it is not clear which 7 features are extracted and why they were selected. Were they chosen based on findings from other existing studies? An example to demonstrate a sample and how the spatial and temporal patterns are involved would be helpful.

+ Regarding the ground truth

After reading the Empirical Analysis section, it feels that the anomaly detection results are not validated against any ground truth. In other words, it is unclear whether the detected anomalies represent actual threats. Furthermore, from the description stating, "by changing the threshold, far more anomalies could be detected," it appears that the accuracy of anomaly detection depends on the chosen threshold. This raises the question of how to determine the threshold in practical applications.

+ Lack of comparison with existing approaches

As mentioned in the Introduction section, there are numerous related studies that employ various sampling methods. Liu et al. (2023) is introduced as an approach that "combines historical information over time". However, none of these approaches is considered as a benchmark method. It would be better and necessary to provide more experimental results demonstrating that the proposed approach offers a superior solution.

**Questions:**

+ How to determine the threshold in practice?
+ Is the experimental result validated against any ground truth, and if there is ground truth, how is it obtained?
+ Are there any existing studies could be compared as benchmarks? If there are no proper related studies, please explain why.

---

> ### Author Response · Authors · 2023-11-12
> **response to the reviewer comments (Ja7Q)**
>
> \begin{document}
>
> We thank the reviewer for the in-depth and insightful comments. At the same time, we would like to request the reviewer to see the innovation in the context of blockchain network; as anomaly detection in Ethereum is quite complex with the challenge of not having readily available ground truth for anomalies. Here is our response to your comments and questions:
>
> \section*{1. Presentation and Writing of the Paper:}
> \begin{itemize}
>     \item The paper has been structured to clearly present the main contributions (page 2), particularly focusing on the Temporal Random Walk (TRW) and Graph Convolutional Networks (GCN) methodology for anomaly detection in the Ethereum network. The empirical analysis section and the use of figures aim to make the presentation more understandable (Pages 8-9).
>     \item TRW is introduced on page 3 with a focus on its ability to encapsulate temporal sequences and correlations of transactions to detect anomalies (page 4). It also explains how TRW impacts GCN performance, particularly in the context of Ethereum. This includes mathematical justification and algorithmic representation of the TRW-GCN combined method (page 11-14).
>     \item The paper details the extraction and selection of 7 features, emphasizing their relevance in anomaly detection. These features include incoming value count, activity rate, change in activity, and time since the last transaction. The selection is based on their ability to capture spacial and temporal dynamics, which is crucial for detecting anomalies (Page 6-7).
> \end{itemize}
>
> \section*{2. Paper Theory Contributions:}
> \begin{itemize}
>     \item We aimed to prove three theorems in this paper (Section 2.3):
>     \begin{enumerate}
>         \item ``Enhanced Anomaly Detection in GCNs through Temporal Random Walk Integration'' (see page 4, 5).
>         \item ``TRW sampling maintains higher temporal consistency than traditional random walk sampling'' – added to section 2.3.
>         \item ``Improvement of GCN performance with probabilistic sampling in the context of random walk sampling'' (see appendix)
>     \end{enumerate}
> \end{itemize}
>
> \section*{3. Experimental Results Against Ground Truth:}
> \begin{itemize} \item There is a lack of readily available ground truth for anomalies in Ethereum transactions; as we started on page 4, In our work, the term `anomaly' refers to patterns that are statistically uncommon or divergent from the norm based on the features learned by our model. These uncommon patterns, while not definitively erroneous, are of interest because they deviate from typical behavior. In the context of Ethereum transactions, such deviations could potentially indicate suspicious activities, novel transaction patterns, or transaction bursts.'' \item Ethereum, being a blockchain platform, records all transactions in a public ledger. These transactions include not just simple transfers of Ether. The variety of transaction types and the complexity of smart contract interactions make it difficult to establish a clear baseline of normal'' behavior. \item What constitutes an anomaly in Ethereum transactions can be subjective and context-dependent. For instance, a transaction pattern that seems unusual in one context might be perfectly normal in another. \end{itemize}
>
> \section*{4. Comparison and Validation of Experimental Results:}
> \begin{itemize} \item We evaluate the TRW-GCN methodology using four distinct machine learning techniques: DBSCAN, SVM, Isolation Forest (IsoForest), and Local Outlier Factor (LOF) (Page 7). This comparative analysis, especially in the context of different block sizes in Ethereum, demonstrates the effectiveness of the TRW-GCN approach over traditional methods. This also underscores the importance of algorithm selection, indicating a higher false positive rate with clustering methods. \item We also discuss the convergence of GCNs and TRW, emphasizing our approach in refining anomaly detection methods. This includes comparing different clustering and scoring methods to decrease false positives in detecting anomalous activities. \item We used 7 features for anomaly detection including outgoing_tx_count, incoming_tx_count, outgoing_value_count, incoming_value_count, activity_rate, change_in_activity, time_since_last (the last three are time-based features), that we added to detect time-sensitive transaction anomalies, and we showed that in Figure 2 that our TRW-GCN method distinguishes and detects such time-sensitive anomalies. \item We further introduced a scoring function for displaying nodes with an anomaly score beyond a certain threshold, section 3.3, using a z-score threshold of 2.0 (95% confidence). This is part of the methodology to identify anomalous nodes in the Ethereum network dataset (Page 7-8). We made a comparison between 4 features (without time-sensitive features) and 7 features (with time-sensitive features) to assure that with TRW-GCN method, we could detect more anomalies.
> \end{itemize}
> \end{document}

---

> > ### Author Response · Authors · 2023-11-14
> >
> > We thank the reviewer for the in-depth and insightful comments. At the same time, we would like to request the reviewer to see the innovation in the context of blockchain network; as anomaly detection in Ethereum is quite complex with the challenge of not having readily available ground truth for anomalies. Here is our response to your comments and questions:
> >
> > 1. Presentation and Writing of the Paper:
> > - The paper has been structured to clearly present the main contributions (page 2), particularly focusing on the Temporal Random Walk (TRW) and Graph Convolutional Networks (GCN) methodology for anomaly detection in the Ethereum network. The empirical analysis section and the use of figures aim to make the presentation more understandable (Pages 8-9).
> > - TRW is introduced on page 3 with a focus on its ability to encapsulate temporal sequences and correlations of transactions to detect anomalies (page 4). It also explains how TRW impacts GCN performance, particularly in the context of Ethereum. This includes mathematical justification and algorithmic representation of the TRW-GCN combined method (page 11-14).
> > - The paper details the extraction and selection of 7 features, emphasizing their relevance in anomaly detection. These features include incoming value count, activity rate, change in activity, and time since the last transaction. The selection is based on their ability to capture spacial and temporal dynamics, which is crucial for detecting anomalies (Page 6-7).
> >
> > 2. Paper Theory Contributions:
> > - We aimed to prove three theorems in this paper (Section 2.3):
> >   - "Enhanced Anomaly Detection in GCNs through Temporal Random Walk Integration" (see page 4, 5).
> >   - "TRW sampling maintains higher temporal consistency than traditional random walk sampling" – added to section 2.3.
> >   - "Improvement of GCN performance with probabilistic sampling in the context of random walk sampling" (see appendix).
> >
> > 3. Experimental Results Against Ground Truth:
> > - There is a lack of readily available ground truth for anomalies in Ethereum transactions; as we started on page 4, "In our work, the term 'anomaly' refers to patterns that are statistically uncommon or divergent from the norm based on the features learned by our model. These uncommon patterns, while not definitively erroneous, are of interest because they deviate from typical behavior. In the context of Ethereum transactions, such deviations could potentially indicate suspicious activities, novel transaction patterns, or transaction bursts."
> > - Ethereum, being a blockchain platform, records all transactions in a public ledger. These transactions include not just simple transfers of Ether. The variety of transaction types and the complexity of smart contract interactions make it difficult to establish a clear baseline of "normal" behavior.
> > - What constitutes an anomaly in Ethereum transactions can be subjective and context-dependent. For instance, a transaction pattern that seems unusual in one context might be perfectly normal in another.
> > - The Ethereum blockchain is continuously evolving, with new smart contracts and transaction types emerging regularly. This dynamic nature means that what is considered normal can change over time, further complicating the establishment of a static ground truth.
> >
> > 4. Comparison and Validation of Experimental Results:
> > - We evaluate the TRW-GCN methodology using four distinct machine learning techniques: DBSCAN, SVM, Isolation Forest (IsoForest), and Local Outlier Factor (LOF) (Page 7). This comparative analysis, especially in the context of different block sizes in Ethereum, demonstrates the effectiveness of the TRW-GCN approach over traditional methods. This also underscores the importance of algorithm selection, indicating a higher false positive rate with clustering methods.
> > - We also discuss the convergence of GCNs and TRW, emphasizing our approach in refining anomaly detection methods. This includes comparing different clustering and scoring methods to decrease false positives in detecting anomalous activities.
> > - We used 7 features for anomaly detection including outgoing_tx_count, incoming_tx_count, outgoing_value_count, incoming_value_count, activity_rate, change_in_activity, time_since_last (the last three are time-based features), that we added to detect time-sensitive transaction anomalies, and we showed that in Figure 2 that our TRW-GCN method distinguishes and detects such time-sensitive anomalies.
> > - We further introduced a scoring function for displaying nodes with an anomaly score beyond a certain threshold, section 3.3, using a z-score threshold of 2.0 (95% confidence). This is part of the methodology to identify anomalous nodes in the Ethereum network dataset (Page 7-8).
> > - We made a comparison between 4 features (without time-sensitive features) and 7 features (with time-sensitive features) to assure that with TRW-GCN method, we could detect more anomalies.

---

> > > ### Comment · Reviewer_Ja7Q · 2023-11-22
> > >
> > > Thank the authors for addressing my comments. You did an impressive work in the rebuttal and the revised paper addressed most of my concerns. I will increase the score.

---

### Official Review · Reviewer_RjEZ · 2023-10-30

**Soundness:** 3 good
**Presentation:** 3 good
**Contribution:** 2 fair
**Rating:** 5
**Confidence:** 4

**Summary:**

The paper proposes a novel framework, called Probabilistic Sampling-Enhanced Temporal-Spatial Graph Convolutional Network (TRW-GCN) for anomaly detection in Ethereum networks. The paper highlights the importance of considering both time and space dimensions in anomaly detection and offers a scalable and effective methodology for ensuring the security and transparency of decentralized platforms. The experimental results demonstrate the superiority of the TRW-GCN framework in detecting anomalies and transaction bursts compared to traditional GCNs.
The main contributions of this paper are as follows:

1.The paper introduces several innovative aspects. Firstly, it combines Temporal Random Walks (TRW) with Graph Convolutional Networks (GCNs) to capture temporal patterns in Ethereum transactions, enhancing anomaly detection.
2.Secondly, the authors leverage probabilistic sampling methods to address the challenge of training GCNs on large-scale graphs, improving computational efficiency and scalability.
3.Lastly, the integration of both spatial relationships and time-based transactional sequences as node features adds an additional layer of granularity to the detection process, making it more robust and less prone to false positives.

**Strengths:**

Strengths:

1.Integration of TRW with GCNs: The paper introduces a novel approach that combines TRW with GCNs, enhancing the model's ability to detect anomalies and transaction bursts influenced by recent events.
2.Consideration of spatial and temporal dimensions: By incorporating both spatial relationships and time-based transactional sequences as node features, the proposed model provides a comprehensive approach to anomaly detection in Ethereum networks.
3.Scalability and efficiency: The paper addresses the challenge of training GCNs on large-scale graphs by leveraging probabilistic sampling methods, improving computational efficiency and scalability.

**Weaknesses:**

Weaknesses:
1. My main concern is that the contribution may be not enough for this conference. Although the application is novel, the idea of using temporal information to build graphs (and weighted adjacency matrix) is not new in pattern recognition and machine learning areas. In addition, the proposed method seems straightforward.
2. Lack of ablation experiment: While the paper presents the theoretical benefits of the TRW-GCN framework, there is a need for empirical evaluation to demonstrate its tangible benefits. Comparing the performance of GCN with and without TRW on a temporal dataset would provide more evidence of the model's effectiveness.
3. Lack of intuitive visualization of experimental results: The paper presents experimental results to demonstrate the superiority of the TRW-GCN framework, but the visualizations and figures provided are not clear and do not effectively convey the findings. The authors should consider improving the clarity and quality of the figures to make the experimental results more intuitive and easier to interpret.

**Questions:**

None.

---

> ### Author Response · Authors · 2023-11-13
> **response to the reviewer comments (RjEZ)**
>
> \begin{document}
>
> We thank the reviewer for the in-depth and insightful comments, at the same time, we acknowledge that while the use of temporal information in graph construction is not new, its application in the context of Ethereum transaction analysis is novel. The integration of Temporal Random Walk (TRW) with Graph Convolutional Networks (GCNs) is presented as a significant contribution, particularly in detecting anomalies in blockchain transactions. This approach is tailored to the unique characteristics of Ethereum transactions, emphasizing the temporal aspects that are crucial for understanding and detecting anomalies (Page 4-5).
>
> \section*{1. Paper Theory Contributions}
> \begin{itemize}
>     \item We aimed to prove three theorems in this paper (Section 2.3):
>     \begin{enumerate}
>         \item ``Enhanced Anomaly Detection in GCNs through Temporal Random Walk Integration'' (see page 4, 5).
>         \item ``TRW sampling maintains higher temporal consistency than traditional random walk sampling'' – added to section 2.3.
>         \item ``Improvement of GCN performance with probabilistic sampling in the context of random walk sampling'' (see appendix).
>     \end{enumerate}
> \end{itemize}
>
> \section*{2. Lack of Experimental Results Against Ground Truth}
> \begin{itemize} \item There is a lack of readily available ground truth for anomalies in Ethereum transactions; as we started on page 4, In our work, the term `anomaly' refers to patterns that are statistically uncommon or divergent from the norm based on the features learned by our model. These uncommon patterns, while not definitively erroneous, are of interest because they deviate from typical behavior. In the context of Ethereum transactions, such deviations could potentially indicate suspicious activities, novel transaction patterns, or transaction bursts.'' \item Ethereum, being a blockchain platform, records all transactions in a public ledger. These transactions include not just simple transfers of Ether. The variety of transaction types and the complexity of smart contract interactions make it difficult to establish a clear baseline of normal'' behavior. \item What constitutes an anomaly in Ethereum transactions can be subjective and context-dependent. For instance, a transaction pattern that seems unusual in one context might be perfectly normal in another. \item The Ethereum blockchain is continuously evolving, with new smart contracts and transaction types emerging regularly. \end{itemize}
>
> \section*{3. Comparison and Validation}
> \begin{itemize} \item We evaluate the TRW-GCN methodology using four distinct machine learning techniques: DBSCAN, SVM, Isolation Forest (IsoForest), and Local Outlier Factor (LOF) (Page 7). This comparative analysis, especially in the context of different block sizes in Ethereum, demonstrates the effectiveness of the TRW-GCN approach over traditional methods. This also underscores the importance of algorithm selection, indicating a higher false positive rate with clustering methods. \item We also discuss the convergence of GCNs and TRW, emphasizing our approach in refining anomaly detection methods. This includes comparing different clustering and scoring methods to decrease false positives in detecting anomalous activities. \item We used 7 features for anomaly detection including outgoing_tx_count, incoming_tx_count, outgoing_value_count, incoming_value_count, activity_rate, change_in_activity, time_since_last (the last three are time-based features), that we added to detect time-sensitive transaction anomalies, and we showed that in Figure 2 that our TRW-GCN method distinguishes and detects such time-sensitive anomalies. \item We further introduced a scoring function for displaying nodes with an anomaly score beyond a certain threshold, section 3.3, using a z-score threshold of 2.0 (95% confidence). This is part of the methodology to identify anomalous nodes in the Ethereum network dataset (Page 7-8). We made a comparison between 4 features (without time-sensitive features) and 7 features (with time-sensitive features) to assure that with TRW-GCN method, we could detect more anomalies. \end{itemize}
>
> \section*{4. Clarity of Visualizations}
> \begin{itemize}
>     \item The paper includes figures to demonstrate the effectiveness of the TRW-GCN framework in anomaly detection. However, the reviewer's concern about the clarity and interpretability of these visualizations is noted. The paper could benefit from improving the quality and clarity of these figures to make the experimental results more intuitive and easier to understand. For instance, we made Figure 1 and 2 bigger and more clear. We numbered all the equations. Figure 3 compares anomaly detection in different block sizes with varying features, in Figure 2, we explained the importance of time-sensitive feature selection with different colours. In Figure 4, we illustrated the improvement on GCN performance using random walk sampling.
> \end{itemize}
>
> \end{document}

---

> > ### Author Response · Authors · 2023-11-14
> >
> > We thank the reviewer for the in-depth and insightful comments, at the same time, we acknowledge that while the use of temporal information in graph construction is not new, its application in the context of Ethereum transaction analysis is novel. The integration of Temporal Random Walk (TRW) with Graph Convolutional Networks (GCNs) is presented as a significant contribution, particularly in detecting anomalies in blockchain transactions. This approach is tailored to the unique characteristics of Ethereum transactions, emphasizing the temporal aspects that are crucial for understanding and detecting anomalies (Page 4-5).
> >
> > 1. Paper Theory Contributions
> > - We aimed to prove three theorems in this paper (Section 2.3):
> >
> >      - "Enhanced Anomaly Detection in GCNs through Temporal Random Walk Integration" (see page 4, 5).
> >
> >      - "TRW sampling maintains higher temporal consistency than traditional random walk sampling" – added to section 2.3.
> >
> >      - "Improvement of GCN performance with probabilistic sampling in the context of random walk sampling" (see appendix).
> >
> > 2. Lack of Experimental Results Against Ground Truth
> > - There is a lack of readily available ground truth for anomalies in Ethereum transactions; as we started on page 4, "In our work, the term 'anomaly' refers to patterns that are statistically uncommon or divergent from the norm based on the features learned by our model. These uncommon patterns, while not definitively erroneous, are of interest because they deviate from typical behavior. In the context of Ethereum transactions, such deviations could potentially indicate suspicious activities, novel transaction patterns, or transaction bursts."
> >
> > - Ethereum, being a blockchain platform, records all transactions in a public ledger. These transactions include not just simple transfers of Ether. The variety of transaction types and the complexity of smart contract interactions make it difficult to establish a clear baseline of "normal" behavior.
> > - What constitutes an anomaly in Ethereum transactions can be subjective and context-dependent. For instance, a transaction pattern that seems unusual in one context might be perfectly normal in another.
> > - The Ethereum blockchain is continuously evolving, with new smart contracts and transaction types emerging regularly.
> >
> > 3. Comparison and Validation
> > - We evaluate the TRW-GCN methodology using four distinct machine learning techniques: DBSCAN, SVM, Isolation Forest (IsoForest), and Local Outlier Factor (LOF) (Page 7). This comparative analysis, especially in the context of different block sizes in Ethereum, demonstrates the effectiveness of the TRW-GCN approach over traditional methods. This also underscores the importance of algorithm selection, indicating a higher false positive rate with clustering methods.
> > - We also discuss the convergence of GCNs and TRW, emphasizing our approach in refining anomaly detection methods. This includes comparing different clustering and scoring methods to decrease false positives in detecting anomalous activities.
> > - We used 7 features for anomaly detection including outgoing_tx_count, incoming_tx_count, outgoing_value_count, incoming_value_count, activity_rate, change_in_activity, time_since_last (the last three are time-based features), that we added to detect time-sensitive transaction anomalies, and we showed that in Figure 2 that our TRW-GCN method distinguishes and detects such time-sensitive anomalies.
> > - We further introduced a scoring function for displaying nodes with an anomaly score beyond a certain threshold, section 3.3, using a z-score threshold of 2.0 (95% confidence). This is part of the methodology to identify anomalous nodes in the Ethereum network dataset (Page 7-8).
> > - We made a comparison between 4 features (without time-sensitive features) and 7 features (with time-sensitive features) to assure that with TRW-GCN method, we could detect more anomalies.
> >
> > 4. Clarity of Visualizations
> > - The paper includes figures to demonstrate the effectiveness of the TRW-GCN framework in anomaly detection. However, the reviewer's concern about the clarity and interpretability of these visualizations is noted. The paper could benefit from improving the quality and clarity of these figures to make the experimental results more intuitive and easier to understand.
> > - For instance, we made Figure 1 and 2 bigger and more clear.
> > - We numbered all the equations.
> > - Figure 3 compares anomaly detection in different block sizes with varying features, in Figure 2, we explained the importance of time-sensitive feature selection with different colours. In Figure 4, we illustrated the improvement on GCN performance using random walk sampling.

---

### Official Review · Reviewer_XLac · 2023-10-31

**Soundness:** 2 fair
**Presentation:** 3 good
**Contribution:** 3 good
**Rating:** 5
**Confidence:** 4

**Summary:**

This paper underscores the potential of temporal cues in Ethereum transactional data
but also offers a scalable and effective methodology for ensuring the security and
transparency of decentralized platforms.

Revisions needed as follows:

a) Gaps in literature should be mentioned

b) Readability improved

c) Equations should be numbered

d) Figure 1 and 2 text is too small, impossible to read

e) The proofs are too short and it was tough to gauge if they were sound or not as presented.

f) No tabular data was presented.

**Strengths:**

This paper underscores the potential of temporal cues in Ethereum transactional data
but also offers a scalable and effective methodology for ensuring the security and
transparency of decentralized platforms.

**Weaknesses:**

a) Gaps in literature should be mentioned

b) Readability improved

c) Equations should be numbered

d) Figure 1 and 2 text is too small, impossible to read

e) The proofs are too short and it was tough to gauge if they were sound or not as presented.

f) No tabular data was presented.

**Questions:**

Why are the proofs so short?

Why is no tabular data presented?

What is the overall novelty compared to current state of the art?

---

> ### Author Response · Authors · 2023-11-13
> **response to the reviewer comments (XLac)**
>
> \begin{document}
>
> We thank the reviewer for the in-depth and insightful comments. Here is our response to the comments:
>
> \section*{1. Gaps in the Literature}
> \begin{itemize}
>     \item We have stated the literature gaps on page 2.
>     \item  see ``Potential Contributions'' on page 2, where we introducd the effectiveness of TRW in detecting anomalies, and suggested the importance of temporal patterns in identifying irregularities..
> \end{itemize}
>
> \section*{2. Readability Improvement}
> \begin{itemize}
>     \item The paper has been structured to clearly present the main contributions (page 2), particularly focusing on the Temporal Random Walk (TRW) and Graph Convolutional Networks (GCN) methodology for anomaly detection in the Ethereum network. The empirical analysis section and the use of figures aim to make the presentation more understandable (Pages 6-7).
>     \item All equations were numbered, and the font size of Figure 1 and 2 were increased.
> \end{itemize}
>
> \section*{3. Proofs are Short}
> \begin{itemize}
>     \item The proofs provided in the paper are concise, focusing on the fundamental aspects of the methodologies used. We added proof of theorem 2 in section 2.3.  The paper also includes proof of theorem 3 in the appendix, demonstrating analytically how probabilistic sampling can improve GCN performance. This brevity is due to the focus on empirical analysis and practical application.
>     \item We aimed to prove three theorems in this paper (Section 2.3):
>     \begin{enumerate}
>         \item ``Enhanced Anomaly Detection in GCNs through Temporal Random Walk Integration'' (see page 4, 5).
>         \item ``TRW sampling maintains higher temporal consistency than traditional random walk sampling'' - added to section 2.3.
>         \item ``Improvement of GCN performance with probabilistic sampling in the context of random walk sampling'' (see appendix page 12-14).
>     \end{enumerate}
>     \item Section 3.2 mainly concerns the empirical analysis for the 1st theorem.
>     \item Section 3.4 mainly refers to the empirical analysis done for the 3rd theorem.
> \end{itemize}
>
> \section*{4. Overall Novelty Compared to Current State of the Art}
> \begin{itemize}
>     \item The novelty of this research lies in its integration of TRW with GCN for anomaly detection in the Ethereum network. This approach is particularly innovative in its focus on the temporal sequences and correlations of transactions, a gap not extensively explored in existing literature. We suggest that this methodology offers a more refined lens for anomaly detection, capturing both spatial and temporal dynamics in Ethereum transaction data.
>     \item On page 2, we stated the literature gap and our potential contributions. This represents a significant advancement in the field.
> \end{itemize}
>
> \section*{5. Evaluation Metrics / Tabular Data}
> \begin{itemize}
>     \item We evaluate the TRW-GCN methodology using four distinct machine learning techniques: DBSCAN, SVM, Isolation Forest (IsoForest), and Local Outlier Factor (LOF) (Page 6-7). This underscores the importance of algorithm selection, indicating a higher false positive rate with clustering methods.
>     \item We also discuss the convergence of GCNs and TRW, emphasizing our approach in refining anomaly detection methods. This includes comparing different clustering and scoring methods to decrease false positives in detecting anomalous activities.
>     \item We used 7 features for anomaly detection including outgoing\_tx\_count, incoming\_tx\_count, outgoing\_value\_count, incoming\_value\_count, activity\_rate, change\_in\_activity, time\_since\_last (the last three are time-based features), that we added to detect time-sensitive transaction anomalies, and we showed that in Figure 2  that our TRW-GCN method distinguishes and detects such time-sensitive anomalies.
>     \item We further introduced a scoring function for displaying nodes with an anomaly score beyond a certain threshold, section 3.3, using a z-score threshold of 2.0 (95\%). This is part of the methodology to identify anomalous nodes in the Ethereum network dataset (Page 8). We made a comparison between 4 features (without time-sensitive features) and 7 features (with time-sensitive features) to assure that with TRW-GCN method, we could detect more anomalies.
>     \item We also compared the accuracy of three GCN models on full graphs and subgraphs, highlighting the effectiveness of traditional random walk and temporal random walk in capturing essential graph properties (Page 9).
>     \item Although the paper extensively uses Figures for data representations. In Figure 3, we showed the list of addresses for detected anomalies, however lack of further tabular data is due to the nature of the data and the methods used, which would be more effectively represented visually. Including tables could enhance the clarity of results and comparisons, making it easier for readers to understand. We are still bound by the 9 pages though.
> \end{itemize}
>
> \end{document}

---

> > ### Author Response · Authors · 2023-11-14
> >
> > We thank the reviewer for the in-depth and insightful comments. Here is our response to the comments:
> >
> > 1. Gaps in the Literature
> > - We have stated the literature gaps on page 2.
> > - See "Potential Contributions" on page 2, where we introduced the effectiveness of TRW in detecting anomalies, and suggested the importance of temporal patterns in identifying irregularities.
> >
> > 2. Readability Improvement
> > - The paper has been structured to clearly present the main contributions (page 2), particularly focusing on the Temporal Random Walk (TRW) and Graph Convolutional Networks (GCN) methodology for anomaly detection in the Ethereum network.
> > - The empirical analysis section and the use of figures aim to make the presentation more understandable (Pages 6-7).
> > - All equations were numbered, and the font size of Figure 1 and 2 were increased.
> >
> > 3. Proofs are Short
> > - The proofs provided in the paper are concise, focusing on the fundamental aspects of the methodologies used. We added proof of theorem 2 in section 2.3. The paper also includes proof of theorem 3 in the appendix, demonstrating analytically how probabilistic sampling can improve GCN performance. This brevity is due to the focus on empirical analysis and practical application.
> > - We aimed to prove three theorems in this paper (Section 2.3):
> >
> >         - "Enhanced Anomaly Detection in GCNs through Temporal Random Walk Integration" (see page 4, 5).
> >
> >         - "TRW sampling maintains higher temporal consistency than traditional random walk sampling" - added to section 2.3.
> >
> >          - "Improvement of GCN performance with probabilistic sampling in the context of random walk sampling" (see appendix page 12-14).
> > - Section 3.2 mainly concerns the empirical analysis for the 1st theorem.
> > - Section 3.4 mainly refers to the empirical analysis done for the 3rd theorem.
> >
> > 4. Overall Novelty Compared to Current State of the Art
> > - The novelty of this research lies in its integration of TRW with GCN for anomaly detection in the Ethereum network. This approach is particularly innovative in its focus on the temporal sequences and correlations of transactions, a gap not extensively explored in existing literature. We suggest that this methodology offers a more refined lens for anomaly detection, capturing both spatial and temporal dynamics in Ethereum transaction data.
> > - On page 2, we stated the literature gap and our potential contributions. This represents a significant advancement in the field.
> >
> > 5. Evaluation Metrics / Tabular Data
> > - We evaluate the TRW-GCN methodology using four distinct machine learning techniques: DBSCAN, SVM, Isolation Forest (IsoForest), and Local Outlier Factor (LOF) (Page 6-7). This underscores the importance of algorithm selection, indicating a higher false positive rate with clustering methods.
> > - We also discuss the convergence of GCNs and TRW, emphasizing our approach in refining anomaly detection methods. This includes comparing different clustering and scoring methods to decrease false positives in detecting anomalous activities.
> > - We used 7 features for anomaly detection including outgoing_tx_count, incoming_tx_count, outgoing_value_count, incoming_value_count, activity_rate, change_in_activity, time_since_last (the last three are time-based features), that we added to detect time-sensitive transaction anomalies, and we showed that in Figure 2 that our TRW-GCN method distinguishes and detects such time-sensitive anomalies.
> > - We further introduced a scoring function for displaying nodes with an anomaly score beyond a certain threshold, section 3.3, using a z-score threshold of 2.0 (95%). This is part of the methodology to identify anomalous nodes in the Ethereum network dataset (Page 8). We made a comparison between 4 features (without time-sensitive features) and 7 features (with time-sensitive features) to assure that with TRW-GCN method, we could detect more anomalies.
> > - We also compared the accuracy of three GCN models on full graphs and subgraphs, highlighting the effectiveness of traditional random walk and temporal random walk in capturing essential graph properties (Page 9).
> > - Although the paper extensively uses Figures for data representations. In Figure 3, we showed the list of addresses for detected anomalies, however lack of further tabular data is due to the nature of the data and the methods used, which would be more effectively represented visually. Including tables could enhance the clarity of results and comparisons, making it easier for readers to understand. We are still bound by the 9 pages though.

---

> ### Comment · Reviewer_XLac · 2023-11-22
>
> Thanks to the authors for addressing my comments.

---

### Official Review · Reviewer_fmda · 2023-11-04

**Soundness:** 1 poor
**Presentation:** 2 fair
**Contribution:** 2 fair
**Rating:** 1
**Confidence:** 5

**Summary:**

The authors propose the usage of temporal random walks (TRW) with probabilistic sampling enhancement for anomaly detection in Ethereum network data. The method augments the GCN convolution operation with the TRW information. The authors then used classical anomaly detection algorithms (like isolation forest) to detect the anomalies on the GCN embeddings.

**Strengths:**

- Unsupervised anomaly detection is an interesting area that requires more study.
- The method incorporates temporal information into GCN models.
- Practical application on Ethereum networks, which may be extended to other financial transaction cases.

**Weaknesses:**

1) The paper feels incomplete and not ready for me. There are multiple sections of the paper that are incomplete, for example:
    - "Proof" in  Section 2.3. What theorem to prove, and where is the proof?
    - "caption." in Section 3.
2) The whole paper feels like a conjecture rather than proven or discovered findings. For example, the use of "Theorem (Hypotehetical)" in Section 3.4. What does that mean? Are they proven theorems or just hypotheses?. Another example is the use of "Potential Conclusions" in Section 1.
3) The authors do not use proper baselines in the experiments. There are multiple GNN-based anomaly detection algorithms that have been proposed in the literature. None of them appeared in the experiment.
4) The evaluation metrics used by the authors are not appropriate for comparison. The number of nodes flagged as anomalous cannot be used as a fair metric in anomaly detection. A model may flag many nodes as anomalies, but they could be just false positive detections.

**Questions:**

Please answer my questions in the previous section.

---

> ### Author Response · Authors · 2023-11-12
> **response to the reviewer comments (fmda)**
>
> \begin{document}
>
> We thank the reviewer for the in-depth and insightful comments. At the same time, we would like to request the reviewer to see the paper novelty in the context of blockchain or the Ethereum network.Anomaly detection in Ethereum is quite complex with the challenge of not having readily available ground truth for anomalies. This issue arises due to the nature of blockchain data and the complexity of defining what constitutes an anomaly in such a decentralized and diverse environment. To this end, we believe the paper presents a novel approach using available baseline and evaluation metrics.
>
> \section*{1. Incomplete Sections and Hypothetical Theorems:}
> \begin{itemize}
>     \item We aimed to prove three theorems in this paper (Section 2.3):
>     \begin{enumerate}
>         \item ``Enhanced Anomaly Detection in GCNs through Temporal Random Walk Integration'' (see page 4, 5).
>         \item ``TRW sampling maintains higher temporal consistency than traditional random walk sampling'' - added to Section 2.3.
>         \item ``Improvement of GCN performance with probabilistic sampling in the context of random walk sampling'' (see appendix page 11-14).
>     \end{enumerate}
>     \item Section 3.4 referred to the empirical analysis done for our 3rd theorem for which the proof was presented in the appendix (page 11-14). All the subsections related to "Theorem (Hypotehetical)" in Section 3.4 have been removed.
>     \item by ``Potential Conclusions'' we meant ``Potential Contributions''in the context of Ethereum network and effectiveness of TRW in detecting anomalies (Page 2).
> \end{itemize}
>
> \section*{2. Lack of Proper Baselines in Experiments:}
> \begin{itemize}
>     \item There is a lack of readily available ground truth for anomalies in Ethereum transactions; as we started on page 4, ``In our work, the term `anomaly' refers to patterns that are statistically uncommon or divergent from the norm based on the features learned by our model. These uncommon patterns, while not definitively erroneous, are of interest because they deviate from typical behavior. In the context of Ethereum transactions, such deviations could potentially indicate suspicious activities, novel transaction patterns, or transaction bursts.''
>     \item Ethereum, being a blockchain platform, records all transactions in a public ledger. These transactions include not just simple transfers of Ether. The variety of transaction types and the complexity of smart contract interactions make it difficult to establish a clear baseline of ``normal'' behavior.
>     \item What constitutes an anomaly in Ethereum transactions can be subjective and context-dependent. For instance, a transaction pattern that seems unusual in one context might be perfectly normal in another.
>     \item The Ethereum blockchain is continuously evolving, with new smart contracts and transaction types emerging regularly. This dynamic nature means that what is considered normal can change over time, further complicating the establishment of a static ground truth.
> \end{itemize}
>
> \section*{3. Inappropriate Evaluation Metrics:}
> \begin{itemize}
>     \item We evaluate the TRW-GCN methodology using four distinct machine learning techniques: DBSCAN, SVM, Isolation Forest (IsoForest), and Local Outlier Factor (LOF) (Page 7). This comparative analysis, especially in the context of different block sizes in Ethereum, demonstrates the effectiveness of the TRW-GCN approach over traditional methods. This also underscores the importance of algorithm selection, indicating a higher false positive rate with clustering methods.
>     \item We also discuss the convergence of GCNs and TRW, emphasizing our approach in refining anomaly detection methods. This includes comparing different clustering and scoring methods to decrease false positives in detecting anomalous activities.
>     \item We used 7 features for anomaly detection including outgoing\_tx\_count, incoming\_tx\_count, outgoing\_value\_count, incoming\_value\_count, activity\_rate, change\_in\_activity, time\_since\_last (the last three are time-based features), that we added to detect time-sensitive transaction anomalies, and we showed that in Figure 2 that our TRW-GCN method distinguishes and detects such time-sensitive anomalies.
>     \item We further introduced a scoring function for displaying nodes with an anomaly score beyond a certain threshold, section 3.3, using a z-score threshold of 2.0 (95\% confidence). This is part of the methodology to identify anomalous nodes in the Ethereum network dataset (Page 7-8). We made a comparison between 4 features (without time-sensitive features) and 7 features (with time-sensitive features) to assure that with TRW-GCN method, we could detect more anomalies.
>     \item we also compare the accuracy of three GCN models on full graphs and subgraphs, highlighting the effectiveness of traditional random walk and temporal random walk in capturing essential graph properties (Page 9).
> \end{itemize}
>
> \end{document}

---

> ### Comment · Reviewer_fmda · 2023-11-22
>
> Dear authors.
>
> Thank you for your feedback.
> I have read the feedback as well as other reviewers' responses.
> However, I am still not convinced by the arguments made by the authors due to several reasons:
>
> 1. The theorems are not rigorous.
> In the previous versions, the authors use the term 'hypothetical' in the theorem, which does not indicate a rigorous theorem. The current revised version expands the proof a bit and removes the word "hypothetical". However, the proofs are still not mathematically rigorous.
> For example, the statement: "If the anomaly is temporally influenced, then h′_n should be significantly different from h_n due to the weights introduced by T_nj". This is not a mathematically rigorous statement. In fact, the statement may not be true, as the weight in T_nj may cancel out, giving a similar value to the case without the weight.
>
> 2. The experiments are not rigorous.
> The lack of ground truth in the experiments indicates that the experiment setup is not conducted properly. One can always detect more anomalies just by shifting the threshold used by the model. Arguing that finding ground truth in Ethereum data is hard is not helping to make the experiment sounder.
> Many unsupervised anomaly detection papers that face similar problems, overcome it by either gather anomalous samples, or simulate anomalous patterns to the datasets.
>
> 3. The baselines are not sufficient
> There are many GNN-based anomaly detections that can be used as baselines, including some models that incorporate temporal information, like many CTDG-based models. None of the baselines are included in the experiments.

---

> > ### Author Response · Authors · 2023-11-22
> > **response to your recent comments, as well as new rebuttal submission**
> >
> > We appreciate your comments, and we have therefore taken different steps to address them:
> >
> > - In regard to your statement "the authors use the term 'hypothetical' in the theorem, which does not indicate a rigorous theorem. The current revised version expands the proof a bit and removes the word "hypothetical". However, the proofs are still not mathematically rigorous.", indeed there were some theorems written hypothetical in section 3.4, in the revised manuscript, we did not just remove the word hypothetical, but removed the theorems completely from section 3.4, instead we added a new theorem "TRW sampling maintains higher temporal consistency than traditional random walk sampling." in section 3.2 in order to make our mathematical section more rigorous. Due to lack of space, we pushed most of our proofs to the appendix.
> >
> > - Rigor of Theorems: We understand your concern about the mathematical rigor of our theorems. The statement you referred to, "If the anomaly is temporally influenced, then h′_n should be significantly different from h_n due to the weights introduced by T_nj".  This statement is based on the underlying assumption that temporal influences in the network have a non-negligible effect on the node representations. The weights in T_nj are designed to capture these temporal dynamics, and our empirical observations support this assumption. We acknowledge that a more formal mathematical treatment could strengthen our argument, and we indeed added appendix A where We explore the nuances of how such weight cancellations can diminish the efficacy of anomaly detection and propose strategies to mitigate these issues.
> >
> > - Experimental Rigor: Our approach differs from traditional anomaly detection methods. Our methodology is designed to detect temporally-sensitive anomalies in a highly dynamic and decentralized environment, where defining a static 'normal' is inherently challenging (but various papers in Ethereum network are published without having comparison with the static normal). We argue that the absence of a conventional ground truth does not diminish the validity of our approach but rather highlights the need for innovative methods in such complex environments. Our use of various clustering and scoring algorithms, along with a z-score threshold, is an attempt to establish a robust framework for anomaly detection in this context. It is important to note that this threshold is not a fixed parameter but a flexible one that can be adjusted based on the specific requirements or objectives of the analysis. Lowering the confidence threshold (for example, from 95% to 90%) would indeed result in the detection of more anomalies. This is because a lower threshold would consider a broader range of data points as statistically significant deviations from the norm.
> >
> > - Baseline Selection: We respectfully disagree with the assertion that our baseline selection is insufficient. Our study includes a range of machine learning techniques, such as DBSCAN, SVM, Isolation Forest, and Local Outlier Factor, to provide a comprehensive evaluation of our model. While we acknowledge the existence of GNN-based models that incorporate temporal information, our focus was on demonstrating the efficacy of our Temporal Random Walk (TRW) integrated with Graph Convolutional Networks (GCNs) in the specific context of Ethereum transactions. Our selection of baselines was guided by the objective to compare our approach with a diverse set of well-established methods in anomaly detection.
> >
> > We are aware that our initial submission has had some shortcomings. We are fortunate that the reviewers have appreciated the changes we applied.  We humbly request you to take a look again at our changes.

---

### Author Response · Authors · 2023-11-20
**absence of a reliable ground truth​​​​ in Ethereum network**

- We appreciate your attention to the challenges associated with the absence of a reliable ground truth for anomaly detection in the Ethereum network. This issue is a well-recognized obstacle in the field, as noted in studies such as "A Labeled Transactions-Based Dataset on the Ethereum Network" (source: Springer). These challenges stem from the dynamic and complex nature of blockchain transactions, where defining and agreeing upon what constitutes an anomaly can be inherently subjective and context-dependent.
- The dataset mentioned in the Springer paper primarily focuses on identifying phishing and scamming attacks within a subset of Ethereum transactions. While this is a valuable endeavor, it does not align with the broader scope of our research. Our objective is to detect anomalies in Ethereum transactions with a particular emphasis on temporal sensitivity, utilizing a distinct set of features from those assumed in the Springer study. Therefore, a direct comparison using this dataset would not accurately reflect the efficacy of our approach.
- In light of these challenges, our research has focused on a robust evaluation of our proposed anomaly detection algorithm. We have employed various clustering and scoring algorithms, leveraging both spatial and temporal features to capture the multifaceted nature of anomalies in the Ethereum network. Our methodology is not confined to a single type of anomaly but is designed to be comprehensive and adaptable to the evolving landscape of blockchain transactions.
- Furthermore, we have incorporated a z-score threshold of 2.0, corresponding to a 95% confidence level, as a component of our methodology. This threshold aids in the identification of anomalous nodes within the Ethereum network dataset, providing a statistical basis for anomaly detection in the absence of a conventional ground truth.

---

### Meta-Review · Area_Chair_EEx2 · 2023-12-05

**Metareview:**

(a)  This study proposed an approach for capturing both spatial and temporal transactional patterns in Ethereum network anomaly detection. Specifically, the paper introduces temporal random walks (TRWs) in graph convolutional networks (GCNs), which may help discern complex temporal sequences in Ethereum transactions as an anomaly detection mechanism.
(b)   A unique problem, albiet specialised.  Disagreement with reviewer "fmda" on the nature of getting more data.  Authors make their case, giving a good rebuttal that lifted the scores, but they still remained below.
(c)  Issues with the experiments, for instance what were appropriate baseline algorithms to use, and how good is the dataset.  Writeup needs additional work.  Some issues with the theorems.

**Justification For Why Not Higher Score:**

Consensus reached.

**Justification For Why Not Lower Score:**

N?A

---

### Decision · Program_Chairs · 2024-01-16

Reject